# Strategic Significance of Low Viral Load of Human Papillomavirus in Uterine Cervical Cytology Specimens

**DOI:** 10.3390/diagnostics12081855

**Published:** 2022-07-31

**Authors:** Nora Jee-Young Park, Claire Su-Yeon Park, Ji Yun Jeong, Moonsik Kim, Su Hyun Yoo, Gun Oh Chong, Dae Gy Hong, Ji Young Park

**Affiliations:** 1Department of Pathology, School of Medicine, Kyungpook National University, Kyungpook National University Chilgok Hospital, Daegu 41404, Korea; pathpjy@knu.ac.kr (N.J.-Y.P.); jjiyun@gmail.com (J.Y.J.); teiroa83@gmail.com (M.K.); 2Clinical Omics Research Center, School of Medicine, Kyungpook National University, Daegu 41405, Korea; gochong@knu.ac.kr; 3KNU Convergence Educational Program of Biomedical Sciences for Creative Future Talents, Department of Biomedical Science, Kyungpook National University, Daegu 41566, Korea; 4Faculty of Nursing, University of Alberta, Edmonton, AB T6G 2B7, Canada; clairesuyeonpark@gmail.com; 5Department of Pathology, National Police Hospital, Seoul 05715, Korea; sue-814@hanmail.net; 6Department of Obstetrics and Gynecology, School of Medicine, Kyungpook National University, Kyungpook National University Chilgok Hospital, Daegu 41404, Korea; chssa0220@hanmail.net

**Keywords:** Human Papillomavirus, HPV DNA, viral load, cervical cytology, shared decision making

## Abstract

Infection with high-risk (HR) Human Papillomavirus (HPV) is associated with the development of precancerous lesions or invasive carcinoma of the uterine cervix. Thus, the high viral load (VL) of HR-HPV DNA currently serves as a representative quantitative marker for cervical cancer. However, the clinical significance of low HPV DNA VL remains undetermined. This study aimed to evaluate the clinical association between the low HPV DNA VL and cytology/histologic diagnosis of cervical samples. We searched the electronic medical databases for the resultant analyses of HPV genotyping among patients who underwent treatment for any cervical lesion or who had undergone gynecological examinations with any positive HPV results according to the national cancer screening service between 2015 and 2016. HPV testing with genotyping and semi-quantitative VL measurement was conducted using an Anyplex^TM^ II H28 Detection assay (H28 assay, Seegene, Seoul, Republic of Korea). The H28 assay is a multiplex semi-quantitative real-time PCR test using the tagging of oligonucleotide cleavage and extension (TOCE) technology. The VL was semi-quantified as high (3+; positive signal before 31 PCR cycles), intermediate (2+; positive between 31 and 39 PCR cycles), or low (1+; positive after 40 PCR cycles). Out of 5940 HPV VL analyses, 356 assays (5.99%) were reported as low VL (1+) of HPV DNA. Matched cytology diagnoses were mostly negative findings (n = 347, 97.5%), except for seven cases of atypical squamous cells of undetermined significance (1.9%) and two cases of atypical glandular cells (0.6%). During the follow-up periods, abnormal cytologic diagnoses were identified, including one case of high-grade squamous intraepithelial lesion (HSIL) and two low-grade squamous intraepithelial lesions (LSILs). The matched, confirmative histologic diagnosis of HSIL cytology was compatible with chronic inflammation, wherein the two LSILs had regular check-ups. None revealed clinically concerned outcomes associated with HPV-related squamous lesions. The cytology was most likely negative for malignancy when the VL of HPV DNA was low (1+). Additional strategic monitoring and management may thus be unnecessary.

## 1. Introduction

Worldwide, cervical cancer is the fourth most common cancer and the fourth leading cause of cancer death in women, with an estimated 604,000 new cases and 342,000 deaths in 2020 [1]. Cervical cancer is by far the most common Human Papillomavirus (HPV)-related disease; nearly all cases of cervical cancer (more than 95%) can be attributed to HPV infection.

More than 200 different HPV types have been listed by the International HPV Reference Center (www.hpvcenter.se; accessed on 20 May 2022), and this number continues to expand [2]. HPV16, 18, 31, 33, 35, 39, 45, 51, 52, 56, 58, and 59, referred to as high-risk (HR) HPV types, have been classified as carcinogenic (IARC Group 1), and other HPV types, HPV26, 53, 66, 67, 68, 70, 73, and 82, have been classified as probably or possibly carcinogenic (IARC Groups 2A and 2B) [3,4]. HR-HPV types are the etiological agents of several cancers, such as those of the cervix, vagina, vulva, anus, penis, and oropharyngeal cancer [5]. HPV6 and 11, represented as LR-HPV types, induce benign genital warts or condyloma acuminata [6].

Notably, most cervical cancer (more than 90%) is due to HR-HPV [1]. Integration of HPV into the host genome is reported in a majority of cervical cancers (83%) [7]. HPV16 can be present as an episomal or integrated form, or both [8], whereas HPV18 has been found integrated into almost all cervical cancers [9]. HR-HPV types, i.e., HPV16 and 18, showed that both oncoproteins E6 and E7 play a key role in cervical cancer by altering pathways involved in the host immune response to establish persistent infection and promote cellular transformation [9].

As an effective screening test for cervical cancer, cervical cytology has so far played a key role in reducing the incidence of cervical cancer in countries with organized screening [10,11,12,13,14]; however, it has limitations in terms of test sensitivity, specificity, encouraging participation in the screening program, etc. In particular, the HPV DNA test has further contributed to a reduction in the incidence of cervical cancer [4,15], which highlights its importance for primary screening and management of cervical cancer [16,17].

To date, the viral load (VL) of HPV has been proposed as a potent discriminator of significance from insignificant HPV infections [18,19]. Several research studies have shown high VL of HPV to be an alternate indicator for persistent HPV infection and a significant predictor of the risk of squamous intraepithelial lesions (SILs) [20,21]; however, debates remain on this issue. Other studies using quantitative real-time polymerase chain reaction (qRT-PCR) targeting HPV16 (E6 and L1 genes) have shown mixed results with respect to the association between mean viral burden and grade of SIL [22,23]. The opposite results have also been reported, which show no relevance between the incidence of cervical lesions and HPV VL [24,25].

The VL of HPV can be measured by various methods, including semi-quantitative (e.g., Hybrid Capture 2) or quantitative polymerase chain reaction (PCR) [26,27]. Most recently, a droplet digital PCR (ddPCR) method has been developed to detect and quantify HPV DNA from various HPV types simultaneously. Rotondo et al. showed that the ddPCR method had high sensitivity, accuracy, specificity, and reproducibility in quantifying HPV DNA sequences [28].

In recent practice, a qRT-PCR method is commonly used to estimate the amount of HPV DNA. The Anyplex^TM^ II HPV28 Detection assay (H28 assay, Seegene, Seoul, Republic of Korea) is a multiplex qRT-PCR test using the tagging of oligonucleotide cleavage and extension (TOCE) technology for simultaneous detection and genotyping of 19 high-risk (HR) HPV types (HPV16, 18, 26, 31, 33, 35, 39, 45, 51, 52, 53, 56, 58, 59, 66, 68, 69, 73, and 82), and 9 low-risk (LR) HPV types (HPV6, 11, 40, 42, 43, 44, 54, 61, and 70) in a single reaction (http://www.seegene.com/assays; accessed on 20 May 2022). The H28 assay also estimates the VL through a repeated melting temperature analysis during the TOCE reaction and reports a semi-quantitative system as low (1+; positive for 40 PCR cycles), intermediate (2+; positive within 31–39 cycles), or high (3+; positive before 31 cycles) [29]. 

Meanwhile, unlike the high VL of HPV, there has been little research on the clinicopathologic significance of low (+) VL of HPV based on H28 assay in cervical lesions. It is not surprising that the clinical association between low HPV VL and cytology/histologic diagnosis of cervical samples remains undetermined, mainly because of the over-sensitivity and inconsistent reproducibility of the test. Such a little-known characteristic may impede gynecologic physicians’ and patients’ best shared decision making for therapeutic plans and consequently bring about unexpected problems, such as excessive levels of patient-perceived anxiety or unnecessary waste of time and money. Therefore, a clear association between low HPV VL and cytology/histologic diagnosis of cervical samples using H28 assay needs to be investigated thoroughly.

In this study, we aimed to evaluate the clinicopathologic significance of an HPV infection with a low (1+) viral DNA load on cervical/vaginal cytology samples obtained from patients with a clinical follow-up due to previous cervical lesions, such as SIL or invasive cervical carcinoma, and from patients who have shown a positive result for HPV in an earlier HPV examination.

## 2. Materials and Methods

### 2.1. Case Selection

We searched the electronic medical database of the Department of Pathology of Kyungpook National University Chilgok Hospital in the Republic of Korea for HPV genotyping analyses among patients with any previous cervical lesions or who had undergone gynecological examinations according to the National Cancer Screening Services, between January 2015 and December 2016 at Kyungpook National University Chilgok Hospital.

A total of 5940 HPV analysis results were obtained. Each of the matched clinical characteristics included age and HPV status at the initial cytology/histopathologic diagnosis plus the follow-up information for the subsequent cytology/histopathologic diagnosis and HPV status. A flowchart shows the sample selection and analysis process in this study (Figure 1).

### 2.2. Anyplex^TM^ II HPV H28 Detection Assay

Briefly, liquid-based cytology (LBC) samples were collected using broom-like devices with detachable heads (BD SurePathTM collection vial, Becton, Dickinson and Company, BD Life Sciences-Integrated Diagnostic Solutions, Sparks, MD, USA) according to the manufacturer’s instructions. The cervical broom was placed into the cervical canal and was rotated 360 degrees around the entire cervical canal. The detachable head was placed in a vial with preservative fluid. The LBC sample collection and transient storage temperature followed the manufacturer’s instructions.

Subsequently, the HPV test with Anyplex HPV28 assay using LBC specimens was performed according to the manufacturers’ instructions. DNA from each LBC sample was extracted using STARMag 96 X4 Universal Cartridge Kit (Seegene, Seoul, Korea). HPV genotyping was conducted using a multiplex polymerase chain reaction (PCR) using a CFX96 PCR Thermal Cycler (Bio-Rad, Hercules, CA, USA), according to manufacturers’ guidelines, using 5 μL of template DNA in a total volume of 20 μL for Anyplex™ II HPV28.

At the initial examination and at least once during a follow-up visit, HPV testing with genotyping and semi-quantitative VL measurement were conducted according to the manufacturer’s instructions using the Anyplex^TM^ II H28 Detection kit (Cat No. HP7S00X, Seegene, Seoul, Korea).

### 2.3. Statistical Analyses

We performed a double-blind test for the cytology/histopathology diagnosis and the HPV status using the Anyplex^TM^ II H28 assay. Descriptive statistical analysis using SPSS Statistics software for Windows, version 21.0 (IBM Corporation, Armonk, NY, USA) was only conducted for the demographic data, including age and follow-up.

## 3. Results

At a baseline examination for the cytology and HPV typing, the mean age was 48.1 (±11.5, standard deviation) years old. The mean follow-up period was 58.3 ± 12.2 months.

Out of 5940 HPV analyses, 356 assays (5.99%) were reported as a low (1+) VL of HPV. Of 356 cases with low (1+) VL of HPV, 150 patients had a previous confirmative diagnosis of HPV-related epithelial lesion. The results are summarized in Table 1.

From the HPV test at the starting time (baseline) of this study, 233 (65.4%) out of the 356 cases were HR-HPV genotypes (Table 2). However, the baseline cytology showed mostly negative findings (n = 347, 97.5%), seven atypical squamous cells of undetermined significance (ASC-US) (1.9%), and two atypical glandular cells (AGC) (0.6%) (Table 3). At that time, three ASC-US cases were determined by a punch biopsy: one case of low-grade squamous epithelial lesion (LSIL) and the other two cases of chronic inflammation. One of the AGC cases was also diagnosed as chronic inflammation on the punch biopsy. Detailed clinical characteristics of abnormal baseline cytology are shown in Table 4.

During the follow-up periods, 263 (73.9%) of 356 patients with a low VL of HPV DNA underwent at least one repeated HPV test. Almost all (97.3%) showed a negative HPV result at the initial repeated test, while the remaining patients (n = 7) were finally negative for HPV at the subsequent tests (Figure 2). Furthermore, three patients were identified to have abnormal cytology: one high-grade squamous epithelial lesion (HSIL) and two LSILs. The patient with HSIL cytology was confirmed as chronic inflammation, not evident for HSIL, with no immunoreactivity for P16 immunohistochemistry on the punch biopsy. Two patients who reported LSIL cytology were regularly followed. Each of the baseline HPV types was HPV53 (HR) for HSIL cytology, and HPV59 (HR) and HPV54 (LR) for LSIL cytology. None revealed medically concerned outcomes associated with HPV-related epithelial lesions during the follow-up periods.

## 4. Discussion

HR-HPV genomic integrations into the host genome are associated with a persistent infection [30]. Either acquired genetic or epigenetic alteration based on the viral genome integration could lead to carcinogenesis in the uterine cervix [30,31,32,33]. Therefore, HPV-related premalignant or malignant lesions contain integrated HPV DNA [34,35], extrachromosomal viral DNA [36], or both [37]; for instance, a study analysis from The Cancer Genome Atlas showed that HPV integration occurred in >80% of HPV-related cervical cancers [38].

The recurrence risk of SIL or invasive carcinoma remains higher among women who undergo treatment for primary HPV-related epithelial lesions. For that reason, such patients need to be put under close surveillance and follow-up examination [39]. Given that between 5% and 20% of cases develop a recurrence within three years and have a more than fivefold increased risk of invasive carcinoma than that of the general population [40,41,42,43], it is essential for those patients to have a continuous cervical/vaginal cytology and HPV testing (with genotyping) in the follow-up periods [44]. However, it is worthwhile to note that most of the previous research for the VL of HPV has focused on the significance of a high load of HPV. As a result, the evidence that a high load of HPV is associated with the persistence of HPV DNA [45] and that DNA persistence is a potent indicator of disease clearance or recurrence [46] seems to have more than enough support. Notably, HPV DNA load decreased with time during the follow-up periods, and HR-HPV DNA was significantly reduced from 90% to 20% at six months after treatment [40,47]. All of these suggest that a quantitative analysis for HPV VL could be worthwhile to monitor for treatment effectiveness and disease recurrence.

The Anyplex^TM^ II HPV28 assay is a multiplex semi-quantitative real-time PCR assay using TOCE technology. The viral load is semi-quantified as high (3+; positive signal before 31 PCR cycles), intermediate (2+; positive between 31 and 39 PCR cycles), or low (1+; positive after 40 PCR cycles) [48]. TOCE technology consistently produces the predicted melting temperature profile in singleplex and multiplex assays and distinguishes multiple targets in a single channel in a homogeneous real-time PCR reaction [49]. The sensitivity of TOCE is reported to be comparable to a standard hydrolysis probe reaction, with a limit of detection of one copy/reaction [50]. The assay also detects the internal-control target gene (human β-globin gene) to ensure the quality of prepared nucleic acids [50].

The H28 assay has been validated in recent HPV-related studies. Cornall et al. showed that both Anyplex^TM^ II HPV28 and Anyplex^TM^ II HPV HR assays had superior performances with higher sensitivity than HC2 (*p* < 0.0001) and were concordant with other commercial assays for HR-HPV detection despite presented semi-quantitative results [51]. Furthermore, although the study design was not identical, Lee et al. evaluated performances of the Anyplex^TM^ II HR assay, sharing the same TOCE technology as the Anyplex^TM^ II HPV28 assay, compared to that of Cobas 4800 HPV and HC2. They assessed the precision of the assay by comparing the positive control results from 15 replicates using clinical specimens with low (quantitation cycle, Cq < 37), intermediate (Cq < 32.2), and high (Cq < 26.7) concentrations of HPV DNA in triplicate per run a day on five different days [50]. The precision tests showed 100% detection of three different levels (low, medium, and high concentration) of pooled clinical specimens using the Anyplex^TM^ II HR assay and the Cobas 4800 [50]. Therefore, we could assume that Anyplex^TM^ II tests, including the H28 assay, would be helpful in follow-up testing and patient management by providing genotyping information beyond HPV 16 and HPV 18 positivity [50].

However, few studies have been conducted on the significance of low VL of HPV. Its value has been virtually ignored due to the difficulty in accurate quantification of low VL and (in some cases) the possibility of contamination by sexual partners or another viral status, which may lead to a false-positive reaction of viral DNA for a short term [45]. Such a dearth of reasonable evidence has made it difficult for either physicians or patients to decide or discuss the best practices and strategies for patient-centered management of cervical lesions when confronted with a medical report indicating low VL of HPV. A partial understanding of VL consequences may consequently lead to wasteful use of limited healthcare resources and accordingly impede patient-centered, value-based care delivery [52].

We initiated this study to contribute to medical science by addressing the clinical relevance between the low VL of HPV and the risk of cervical lesions. In this study, none of the patients who reported as low (1+) VL of HPV revealed clinically concerned outcomes associated with HPV-related epithelial lesions during follow-up. Therefore, in accordance with what the study found, we carefully propose that clinicians and patients would have a better shared decision making with simply the monitoring and management of HPV VL, instead of unnecessary additional tests or procedures, when the low (1+) VL of HPV status is detected in an H28 assay.

In particular, there are two further significant implications obtained from this study. Although cervical cancer can develop as long as HPV infection persists, a quantitative contribution to cervical carcinogenesis remains unclear regarding the duration and viral load of HPV infection. Fortunately, our results could provide a practical reference for the clinical treatment of patients. Those with a low VL of HPV should be managed more conservatively and safely or in a step-by-step manner considering familial and social environmental factors, such as pregnancy, rather than immediate management.

The other is that, even though the LR-HPVs are not subject to screening and follow-up for cervical cancer, it is impossible to know whether the viral subtype is from an LR- or HR-HPV group before performing the HPV subtyping test. Moreover, one should be aware of the possibility of cross-reaction or multiple HPV infections. This study demonstrated the clinical significance of the low VL of HPV in practical management. In other words, the cases with a lower VL of LR-HPV, as expected, did not have any precancerous or cancerous lesions during the follow-up; furthermore, in cases of HR-HPV groups, they did show identical findings. Therefore, we proposed that the low VL, regardless of LR- and HR-HPV subtyping, can emphasize the significance of conservative strategies for screening or follow-up of cervical cancer.

It is also obvious that quality control of the laboratory performance is very important for either the primary screening test or a strategic assessment of the post-treatment testing utility for HPV infection. The laboratory performance should adhere to a well-established protocol of intra-laboratory quality assurance and be kept under close review, thereby ensuring a reliable data processing system within the institution. Once an individual laboratory is accredited for HPV molecular testing, intra-institutional strategic guidelines may also be proposed, which ultimately contribute to a standardized decision on “what to do” for the best follow-up care.

This study had several inherent limitations regarding small sample size, sample type, and analytic methods. In detail, first, by selecting patients with HPV infection from one tertiary medical institute in the Republic of Korea, we might have biased our cohorts towards patients with HPV DNA persistence, thus undermining generalization. Second, we only assessed the HPV DNA load using a semi-quantitative measurement by a repeated melting temperature analysis of the HPV28 assay, which may damage validity to some extent. Further studies should propose a clinical cut-off for the viral load for the reliability of HPV testing. Third, the method of HPV28 detection may present inflated positive results due to its excessive sensitivity for the primary HPV screening test. By extension, there remain the possibilities of transient HPV infection or contamination, requiring further efforts to refine the sensitivity of the measurement and increase reliability in future studies. These limitations support a need to verify the study finding through large data obtained from rigorous replication of research in multi-institute settings and the general population. However, we believe that our results outweigh the limitations in that it is the first study to provide at least a minimum of medical evidence on the possibility of strategic utilization of low VL of HPV for monitoring cervical lesions.

## 5. Conclusions

This study showed that none of the patients with a low VL of HPV DNA revealed clinically concerned outcomes associated with HPV-related squamous lesions, regardless of HR- or LR-HPV genotypes. Therefore, we could provide a practical guideline for the patient with a low VL of HPV DNA. When referring to the strategic monitoring and management of patients who undergo treatment for previous HPV-related epithelial lesions and patients who had an HR-HPV detection earlier, those with a low VL of HPV using the AnyplexTM II H28 assay should be managed in a conservative and safe or step-by-step manner considering familial and social environmental factors, such as pregnancy, rather than immediate management. In these particular situations, the accuracy and confidence of laboratory HPV testing need to be consistently guaranteed.

## Figures and Tables

**Figure 1 diagnostics-12-01855-f001:**
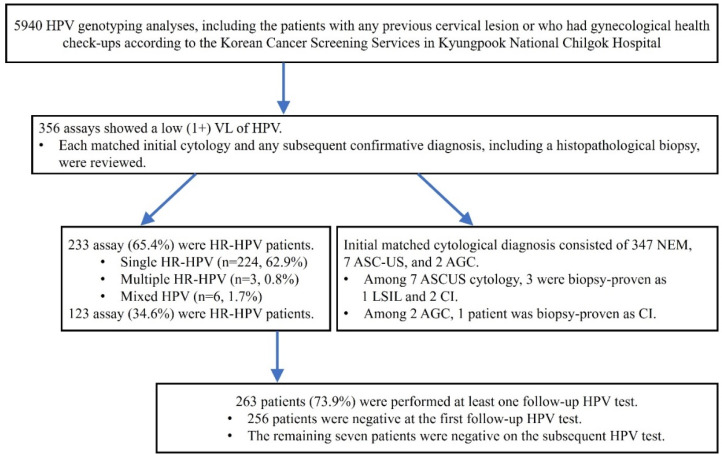
Flowchart of the sample selection and analysis process in this study.

**Figure 2 diagnostics-12-01855-f002:**
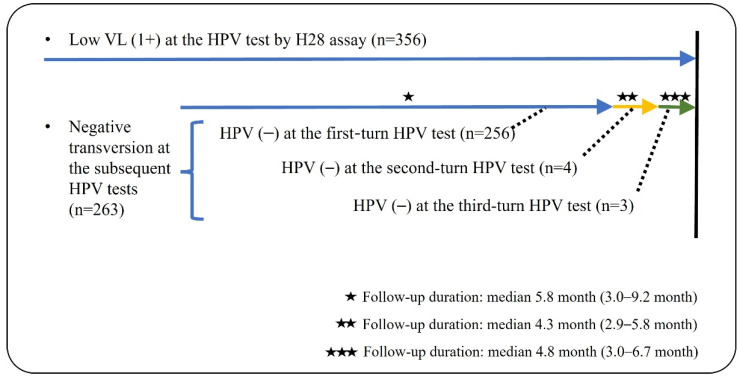
Illustration for clinical follow-ups of low VL (1+) HPV test in this study.

**Table 1 diagnostics-12-01855-t001:** Previous HPV-related lesions of the cases with low (1+) VL of HPV.

Original Site	Previous Histologic Diagnosis	Cases [No.]
Uterus, cervix	Condyloma acuminatum	2
	LSIL (CIN1)	20
	HSIL (CIN2 or CIN3)	18
	Squamous carcinoma in situ	43
	Microinvasive squamous cell carcinoma	8
	Invasive squamous cell carcinoma	30
	Adenocarcinoma in situ	2
	Invasive adenocarcinoma	18
Vagina	HSIL (VAIN2)	1
	Invasive squamous cell carcinoma	1
	Invasive adenocarcinoma	1
	Neuroendocrine carcinoma	1
Vulva	Extramammary Paget disease	1
	Squamous cell carcinoma in situ	1
	Invasive squamous cell carcinoma	3
Total		150

Abbreviations: ASC-US, atypical squamous cell of undetermined significance; AGC, atypical glandular cells; CIN, cervical intraepithelial lesion; HSIL, high-grade squamous intraepithelial lesion; LSIL, low-grade squamous intraepithelial lesions; VAIN, vaginal intraepithelial lesion.

**Table 2 diagnostics-12-01855-t002:** Baseline HPV types in patients with low (1+) VL of HPV.

HPV Genotypes	Single Infection [No. (%)]	Multiple Infection [No. (%)]
HR-HPV	224 (62.9%)	3 (0.8%)
HPV16	20 (5.6%)	
HPV18	10 (2.8%)	
Other HR-types	HPV26 (n = 2, 0.6%) HPV31 (n = 10, 2.8%) HPV33 (n = 3, 0.8%) HPV35 (n = 5, 1.4%) HPV39 (n = 28, 7.9%) HPV45 (n = 2, 0.6%) HPV51 (n = 10, 2.8%) HPV52 (n = 24, 6.7%) HPV53 (n = 21, 5.9%) HPV56 (n = 23, 6.5%) HPV58 (n = 17, 4.8%) HPV59 (n = 9, 2.5%) HPV66 (n = 11, 3.1%) HPV68 (n = 29, 8.1%)	HPV51 + HPV52 (n = 1, 0.3%) HPV53 + HPV68 (n = 1, 0.3%) HPV59 + HPV66 (n = 1, 0.3%)
LR-HPV	115 (32.6%)	8 (2.2%)
HPV6	8 (2.2%)	
HPV11	0 (0%)	
Other LR-types	HPV40 (n = 9, 2.5%) HPV42 (n = 10, 2.8%) HPV43 (n = 10, 2.8%) HPV44 (n = 13, 3.7%) HPV54 (n = 39, 11.0%) HPV61 (n = 18, 5.1%) HPV70 (n = 10, 2.8%)	HPV42 + HPV54 (n = 1, 0.3%) HPV42 + HPV70 (n = 1, 0.3%) HPV44 + HPV54 (n = 1, 0.3%) HPV44 + HPV61 (n = 1, 0.3%) HPV54 + HPV61 (n = 1, 0.3%) HPV61 + HPV70 (n = 3, 0.8%)
Mixed (HR + LR) HPV	–	6 (1.7%)
	–	HPV26 + HPV42 (n = 1, 0.3%) HPV31 + HPV43 (n = 1, 0.3%) HPV35 + HPV54 (n = 1, 0.3%) HPV52 + HPV70 (n = 1, 0.3%) HPV58 + HPV42 (n = 1, 0.3%) HPV59 + HPV54 (n = 1, 0.3%)

Abbreviations: HPV, Human Papillomavirus; HR, high-risk; LR, low-risk.

**Table 3 diagnostics-12-01855-t003:** Baseline cytological diagnosis in patients with low (1+) VL of HPV.

Cytological Diagnosis	HPV (1+)	HR-HPV	LR-HPV
Negative	347	227 (65.4%)	120 (34.6%)
Negative for malignancy	235	160 (68.1%)	75 (31.9%)
Reactive cellular change	46	32 (69.6%)	14 (30.4%)
Atrophy	35	19 (54.3%)	16 (45.7%)
Shift in flora suggestive of bacterial vaginosis	19	12 (63.2%)	7 (36.8%)
Fungal organisms morphologically consistent with Candida spp.	11	4 (36.4%)	7 (63.6%)
Trichomonas vaginalis	1	0 (0%)	1 (100%)
Atypical cells	9	6 (66.7%)	3 (33.3%)
ASC-US	7	4 (57.1%)	3 (42.9%)
AGC	2	2 (100%)	0 (0%)
Total	356 (100%)	233 (65.4%)	123 (34.6%)

Abbreviations: ASC-US, atypical squamous cell of undetermined significance; AGC, atypical glandular cells.

**Table 4 diagnostics-12-01855-t004:** Clinical characteristics of abnormal baseline cytology.

Baseline Cytology	Infection	HPV Type	Related Bx	Follow-Up Cytology
ASC-US	Single	HR (16)	LSIL	Negative
ASC-US	Single	HR (16)	CI	Negative
ASC-US	Single	HR (33)	-	Negative
ASC-US	Single	LR (43)	-	Negative (Atrophy)
ASC-US	Multiple	Mixed (16,44)	CI	Negative
ASC-US	Single	HR (51)	-	Negative
ASC-US	Single	HR (56)	-	Negative
AGC	Single	HR (39)	CI	Negative (Candida spp.)
AGC	Single	HR (56)	-	Negative

Abbreviations: ASC-US, atypical squamous cell of undetermined significance; AGC, atypical glandular cells; Bx, biopsy; CI, chronic inflammation; LSIL, low-grade squamous intraepithelial lesion.

## Data Availability

The data presented in this study are available on request from the corresponding author. The data are not publicly available due to privacy and ethical issue.

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
