# Peer review of "Strategic Significance of Low Viral Load of Human Papillomavirus in Uterine Cervical Cytology Specimens"

_diagnostics, 2022, doi:10.3390/diagnostics12081855_

Round 1

Reviewer 1 Report

Your work concerning the clinical significance of HPV low viral load (VL) is of great matter since most of positive HPV tested women during cervix cancer screening will show viral clearance during the months or years to come and about 90% of them will present normal cytology. Your study concerns women followed in a referral centre in Korea as mentioned in the limitations of this study in Discussion section in page 6 and should go on general population. Furthermore, to evaluate the impact of low VL on clinical outcome, you should also report the outcome from positive women with intermediate and high VL (is that different ?).

Your study is based on a semi-quantitative genotyping test (Seegene H28 assay,) with low VL defined as 1+ and Ct over 40 (added in abstract and method sections). It is stated in the introduction section that inconsistent reproducibility can be observed but you report in the discussion section page 5 a study with this method showing 100% precision using repeated testing with low, intermediate and high VL. However, in this reported study, precision for low viral load was observed for value <37 and not >40, could you extrapolate your results with low VL over 40 from this study ?

Seegene H28 assay is also related to LR and HR HPV types, the authors stated in their discussion page 6 that woment with low VL unregarding LR or HR types are sharing the same clinical outcome (mostly with normal cytology). However, one woman showed HSIL cytology with a HR type (HPV53) wherease two with LSIL had HR and LR types. I understand that these 3 women at the end of results section in page 5 showed « no medically concerned outcomes », is that mean that control cytology or colposcopy was normal ? Nevertheless, LR HPV types should not be included in HPV tests as recommanded for screening (123 among 356 low VL in your study had LR types, 3 among them had ASCUS cytology and got normal, I suppose, during follow-up ?).

Repeated testing is reported in results section page 4 (263 among the 356 low VL), 97.3% of them were negative at the first HPV control test, what is the mean delay from the first test with low VL and the second with negative result ? It is interesting to note if it is a rapid viral clearance. Did you observed further positive test after a negative control (transient clearance) ?

Table 2 should be on the same page. Statistical p-value could be added in table 3 for comparison between LR and HR types (for example were candidosis more frequent in LR types with 7 cases versus 4 cases ?).

Author Response

DATE 07/25/22

Prof. Dr. Andreas Kjaer

Editor-in-Chief

Diagnostics

Dear Editor,

First of all, we sincerely appreciate the assistant editor of Diagnostics, Mr. George Popovici, and the section managing editor, Ms. Kama Chen, for generously allowing me to have very sufficient time in improving this paper, and further providing me additional help for its revision. We would also like to sincerely thank Diagnostics’s reviewers for giving us thoughtful and insightful review comments.

Please find enclosed our revised manuscript entitled “Strategic Significance of Low Viral Load of Human Papillomavirus in Uterine Cervical Cytology Specimens” and a point-by-point response to each reviewer’s comments, which we request you to consider for publication as an original research paper in Diagnostics.

The human papillomavirus (HPV) DNA test has so far contributed to a significant reduction in the incidence of cervical cancer, which highlights its importance for primary screening and management in cervical cancer. The high viral load (VL) of HPV has been proposed as a potent discriminator of its significance distinct from insignificant HPV infections. However, there has been little research on the clinicopathologic significance of low VL of HPV in cervical lesions, unlike the high VL of HPV.

In this study, we sought to evaluate the clinical significance of low VL of HPV and cytology/histologic diagnosis of cervical samples. HPV testing with genotyping and semi-quantification of VL was performed using AnyplexTM II H28 assay. Approximately 6% was reported as low (1+) VL of HPV DNA. When the VL of HPV DNA was low (1+), the cytology was most likely negative for malignancy. Furthermore, none of the patients who reported a low (1+) VL of HPV revealed clinically concerned outcomes associated with HPV-related epithelial lesions during the follow-up periods. Therefore, we carefully propose that clinicians are recommended to have an evidence-based, shared decision-making process with patients in order to prevent patients from experiencing unnecessary, costly additional tests or procedures. We believe that this article is relevant to the scope of your journal and will be of interest to its readership.

The manuscript has been carefully reviewed by an experienced editor whose first language is English and who specializes in editing papers written by scientists whose native language is not English.

We are looking forward to hearing a positive evaluation from Diagnostics.

Sincerely,

Nora Jee-Young Park, M.D., Ph.D.,

Assistant Professor, Department of Pathology, Kyungpook National University Medical Center,

Kyungpook National University, School of Medicine,

807 Hoguk-ro, Buk-gu, Daegu, 41404, Korea

Email: jyparkmd@knu.ac.kr; Phone: +82-53-200-3405; Fax: +82-53-200-3399

Reviewer 2 Report

In this article, Park and colleagues analyze the association between the low HPV viral load and the clinical diagnosis based on cytological or histological result among women with a cervical lesion.

Viral load was evaluated using a commercial assay AnyplexII HPV28 that can detect 28 different HPV genotypes and establish a semi-quantitative analysis of viral load based on a range of Ct value detected during the amplification. Analysis was performed on 5940 samples collected at baseline and 263 samples collected from women with a low viral load at baseline and returned for a follow-up visit.

The objective of this study is very interesting to better understand the role of these HPV infection associated with a low viral and their impact on clinical outcome.

Few results regarding low viral load HPV infection are described in literature and it is important to investigate this field. However, this manuscript has major issues to arrange.

The use of the English language is adequate.

Abstract: Authors should better describe results obtained from analysis performed on samples collected at follow-up visits.

Introduction:

  • Sections regarding HPV genotype description should be better descried.
  • Furthermore, the part describing the method used to determine HPV viral load should be better thorough and reported the importance to have a pre-analytical and analytical methods well defined to have results that could be compared to other data. Moreover, other new techniques such as droplet digital PCR are currently used for absolute viral load quantification, especially for low viral load detection.

Material and Methods:

  • 1. Case selection: Park and colleagues should better specify the population considered for this study and if they performed a calculation to determine the sample size to have statistically significant results. They also should report the rational of selecting just samples collected from women with low viral load for the analysis at follow-up visit. A graph reporting the study population should help to specify this section
  • 2. AnyplexII HPV28 detection: Authors should correctly report the name of assay used through the manuscript. Sample collection and pre-analytical analysis should be reported more in depth. Were the samples stored in a biobank? At which temperature? This data is important because sample defrosting could impact on viral load quantification.

Results:

  • If the aim of this study is to evaluate the clinical association between low HPV viral load and cervical lesions, why authors reported also vaginal and vulvar lesions in Table 1? This should be specified.
  • From the analysis of samples collected from 9 women with cytological lesions, which HPV type was detected? Were these single or multiple HPV infections? Author should describe this result in the text.
  • Data obtained from the analysis of samples collected at follow-up visits are not very well reported. Authors should better specify how many follow-up visits and testing were performed and results obtained at each follow-up.

Discussion:

  • Authors should better describe the limits of this study regarding sample size, type of samples, pre-analytic methods used to obtain the results.

Conclusions:

  • Conclusions should be more associated with these first preliminary results. Moreover, authors should underline that a larger data obtained from future study could give better indications regarding the significance of HPV low viral loads.

References: References should be updated.

Author Response

(The authors gave the same response as above.)

Reviewer 3 Report

The manuscript (ID diagnostics-1667185) entitled “Strategic Significance of Low Viral Load of Human Papilloma-virus in Uterine Cervical Cytology Specimens” by Dr. Jee-Young Park describes an investigation of thethe clinical association between the low amount of HPV DNA load and cytology/histologic diagnosis and prognosis of cervical samples. Main results indicate that out of 5940 samples, 356 (5.99%) were reported as carrying low amount of HPV DNA load. None of the patients who reported as low HPV DNA load revealed clinically concerned outcomes associated with HPV-related epithelial lesions/cervical cancer during follow-up. One of the main strengths of the work is that the HPV load, quantified by a semi-quantitative real-time PCR method (AnyplexTM II H28 Detection assay), was determined in a very large group of clinical samples (n=5940). One of the main limitations of the study is that the PCR method employed is semiquantitative. Thus, a clear indication of the amount of the viral DNA cannot be obtained.  Although several minor modifications can be made for improving the manuscript, this is a well written and concise investigation. The scientific writing style is adequate. Tables are informative and detailed. In general, the ms will improve our knowledge on the prognostic implication of HPV DNA load, in this case low viral load, in cervical cancer and other HPV-related diseases of the female reproductive tract.  Considering the aforementioned aspects, I therefore recommend a minor revision. Only few minor improvements are required:

Comments
1.    I suggest replacing the annotation “HR HPV” and “LR HPV”with “HR-HPV” and LR HPV  throughout the text 
2.    “HPV infection is closely related to the development of cervical cancer”. The majority of cervical cancers are HPV16 positive, while HPV18 is the second most abuntand oncogenic HPV related to this malignancy (DOI 10.3389/fonc.2019.00355).  The difference in terms of oncogenic potential between high and low risk HPVs and their role in cervical cancer, should be more clearly detailed in this sentence and in the following sentences. 
3.    Besides the quantitative real-time polymerase chain reaction (qRT-PCR), the highly sensitive droplet-digital PCR has demonstrated to be highly reliable in detecting low amounts (range from 0.002 to 51.02 copies of viral DNA/cell) of HPV DNA in cervical premalignant lesions (DOI: 10.3389/fmicb.2020.591452). This information should be included
4.    The “viral titer” is ambiguous as it can be confounded with the anti-HPV antibody titer. I suggest replacing “viral titer” with “viral DNA load”
5.    Supporting references should be included in the methods section, especially in the “statistical analyses” sub section
6.    Methods, “The total DNA was extracted and purified from an aliquot of 100μl sample” the exact sample type should be detailed in this section
7.    Have the patients resulted HPV positive with intermediate and high viral DNA amount been excluded? If yes this information (and rates) should be included. 
8.    “During the follow-up periods, of the 356 cases with low VL of HPV, at least one repeated HPV test was performed in 263 patients (73.9%) during the follow-up period.” English should be improved
9.    Discussion, “HR HPV genomic integrations” I would say: HR-HPV genomic integrations into the host genome
10.    Disucsison, the word “assessment” should not be underlined

Author Response

DATE 07/25/22

Prof. Dr. Andreas Kjaer

Editor-in-Chief

Diagnostics

Dear Editor,

First of all, we sincerely appreciate the assistant editor of Diagnostics, Mr. George Popovici, and the section managing editor, Ms. Kama Chen, for generously allowing me to have very sufficient time in improving this paper, and further providing me additional help for its revision. We would also like to sincerely thank Diagnostics’s reviewers for giving us thoughtful and insightful review comments.

Please find enclosed our revised manuscript entitled “Strategic Significance of Low Viral Load of Human Papillomavirus in Uterine Cervical Cytology Specimens” and a point-by-point response to each reviewer’s comments, which we request you to consider for publication as an original research paper in Diagnostics.

The human papillomavirus (HPV) DNA test has so far contributed to a significant reduction in the incidence of cervical cancer, which highlights its importance for primary screening and management in cervical cancer. The high viral load (VL) of HPV has been proposed as a potent discriminator of its significance distinct from insignificant HPV infections. However, there has been little research on the clinicopathologic significance of low VL of HPV in cervical lesions, unlike the high VL of HPV.

In this study, we sought to evaluate the clinical significance of low VL of HPV and cytology/histologic diagnosis of cervical samples. HPV testing with genotyping and semi-quantification of VL was performed using AnyplexTM II H28 assay. Approximately 6% was reported as low (1+) VL of HPV DNA. When the VL of HPV DNA was low (1+), the cytology was most likely negative for malignancy. Furthermore, none of the patients who reported a low (1+) VL of HPV revealed clinically concerned outcomes associated with HPV-related epithelial lesions during the follow-up periods. Therefore, we carefully propose that clinicians are recommended to have an evidence-based, shared decision-making process with patients in order to prevent patients from experiencing unnecessary, costly additional tests or procedures. We believe that this article is relevant to the scope of your journal and will be of interest to its readership.

The manuscript has been carefully reviewed by an experienced editor whose first language is English and who specializes in editing papers written by scientists whose native language is not English.

We are looking forward to hearing a positive evaluation from Diagnostics.

Sincerely,

Nora Jee-Young Park, M.D., Ph.D.,

Assistant Professor, Department of Pathology, Kyungpook National University Medical Center,

Kyungpook National University, School of Medicine,

807 Hoguk-ro, Buk-gu, Daegu, 41404, Korea

Email: jyparkmd@knu.ac.kr; Phone: +82-53-200-3405; Fax: +82-53-200-3399

This manuscript is a resubmission of an earlier submission. The following is a list of the peer review reports and author responses from that submission.

Round 1

Reviewer 1 Report

The presented study is interesting and important in terms of improving the tactics of the approach to treating HPV-positive patients with the risk of developing cervical pathology, as well as the formation of risk groups for this pathology.

However, there are also a number of criticisms for this work:

  1. The section "Introduction" incorrectly presents information on the number of oncogenic types of the virus.
  2. There is no definition for the concepts of high and low viral load. How much DNA from the virus are we talking about?
  3. There is no decryption for "any previous cervical lesion"
  4. For patients with low viral load, there is no complete characterization of clinical and pathological parameters.
  5. It is not specified whether a repeated HPV test was performed to exclude spontaneous viral elimination during the follow-up period.
  6. Continuous persistence of HPV, even with a low viral load for 10 years or more, can lead to the development of dysplastic and tumor changes in the epithelium of the cervix.
  7. The results of statistical received data processing are not provided. SEM scores for mean age (± 11.5) and follow-up period (± 12.2) are not represented correctly
  8. The list of references contains only 14.3% of sources over the past 5 years. It is necessary to update the literature data for more current ones.
  9. The conclusion presented on the results of the work is premature in view of the above.

Reviewer 2 Report

see review enclosed.
